# Low Prevalence of SARS-CoV-2 in Farmed and Free-Ranging White-Tailed Deer in Florida

**DOI:** 10.3390/v16121886

**Published:** 2024-12-06

**Authors:** Savannah G. Grace, Kristen N. Wilson, Rayann Dorleans, Zoe S. White, Ruiyu Pu, Natasha N. Gaudreault, Konner Cool, Juan M. Campos Krauer, Laura E. Franklin, Bambi C. Clemons, Kuttichantran Subramaniam, Juergen A. Richt, John A. Lednicky, Maureen T. Long, Samantha M. Wisely

**Affiliations:** 1Department of Wildlife Ecology and Conservation, University of Florida, Gainesville, FL 32611, USA; savannah.grace@ufl.edu (S.G.G.); knwilson@ufl.edu (K.N.W.); rdorleans@ufl.edu (R.D.); zseganish@ufl.edu (Z.S.W.); laurafranklin@ufl.edu (L.E.F.); 2Emerging Pathogens Institute, University of Florida, Gainesville, FL 32610, USA; pur@ufl.edu (R.P.); kuttichantran@ufl.edu (K.S.); jlednicky@phhp.ufl.edu (J.A.L.); longm@ufl.edu (M.T.L.); 3Department of Diagnostic Medicine/Pathobiology, College of Veterinary Medicine, Kansas State University, Manhattan, KS 66506, USA; nng5757@vet.k-state.edu (N.N.G.); konnerc@vet.k-state.edu (K.C.); jricht@vet.k-state.edu (J.A.R.); 4Department of Large Animal Clinical Sciences, College of Veterinary Medicine, University of Florida, Gainesville, FL 32608, USA; jmcampos@ufl.edu; 5Fish and Wildlife Research Institute, Florida Fish and Wildlife Conservation Commission, Gainesville, FL 32601, USA; bambi.clemons@myfwc.com; 6Department of Infectious Diseases and Immunology, College of Veterinary Medicine, University of Florida, Gainesville, FL 32610, USA; 7Department of Environmental and Global Health, College of Public Health and Health Professions, University of Florida, Gainesville, FL 32610, USA

**Keywords:** SARS-CoV-2, zoonotic transmission, white-tailed deer, Florida

## Abstract

Severe acute respiratory syndrome coronavirus 2 (SARS-CoV-2) has been detected in multiple animal species, including white-tailed deer (WTD), raising concerns about zoonotic transmission, particularly in environments with frequent human interactions. To understand how human exposure influences SARS-CoV-2 infection in WTD, we compared infection and exposure prevalence between farmed and free-ranging deer populations in Florida. We also examined the timing and viral variants in WTD relative to those in Florida’s human population. Between 2020 and 2022, we collected respiratory swabs (N = 366), lung tissue (N = 245), retropharyngeal lymph nodes (N = 491), and serum specimens (N = 381) from 410 farmed and 524 free-ranging WTD. Specimens were analyzed using RT-qPCR for infection and serological assays for exposure. SARS-CoV-2 infection was detected in less than 1% of both northern Florida farmed (0.85%) and free-ranging (0.76%) WTD. No farmed deer possessed virus-neutralizing antibodies, while one free-ranging WTD tested positive for SARS-CoV-2 antibodies (3.45%). Viral sequences in infected WTD matched peaks in human cases and circulating variants, indicating human-to-deer spillover but at a lower frequency than reported elsewhere. Our findings suggest a reduced risk of SARS-CoV-2 spillover to WTD in northern Florida compared to other regions, highlighting the need for further research on transmission dynamics across North America.

## 1. Introduction

Severe acute respiratory virus 2 (SARS-CoV-2), the causative agent of coronavirus disease 2019 (COVID-19), is the pathogen that initiated the global pandemic declared by the World Health Organization (WHO) in 2020. Multiple mammalian species, including certain cervids, are susceptible to SARS-CoV-2 infection [1,2,3,4,5]. White-tailed deer (*Odocoileus virginianus*) (WTD) are of particular concern for potential anthropozoonosis due to their widespread distribution, dense populations, synanthropic nature, genetic similarity of their angiotensin-converting enzyme 2 (ACE2) protein-binding motif to that of humans [6], and demonstrated ability to transmit SARS-CoV-2 from deer-to-deer [7,8].

Surveillance studies in North America have detected a high prevalence of either viral RNA or specific anti-SARS-CoV-2 antibodies in both farmed and free-ranging WTD [1,2,6,8,9,10,11,12,13,14,15,16,17,18,19]. Diverse viral lineages have been identified in free-ranging WTD populations across multiple states of the United States, including sublineages of variants of concern such as Alpha [14,16,17], Gamma [14,17], Delta [14,16,17], and Omicron [14]. Many of the isolates were genetically similar to those circulating in nearby human populations at the time of identification, suggesting that spillover from humans to deer had occurred.

Evidence of spillover raises concerns about deer becoming reservoir hosts and a source for spillback of the virus into humans, the risk of which may be increased in situations where high densities of susceptible animals are in close contact with humans. For example, zoonotic transmission and spillback into humans were documented in connection with outbreaks of SARS-CoV-2 in farmed mink in both Europe and North America [20,21]. In Denmark, these transmission dynamics resulted in a government-mandated culling of all farmed mink in 2020.

In the United States, cervid farming has become a billion-dollar industry. There are over 10,000 cervid farming facilities across the country and more than 300 in Florida alone, many of which primarily breed WTD [22]. Farmed deer in Florida are raised in close proximity to humans and can be stocked at a density that is approximately 12 times higher than that of wild deer [23], which can increase the risk of SARS-CoV-2 transmission among deer and between humans and deer. Certain farming practices, such as bottle feeding and administering medical treatments, may also contribute to the transmission of the virus. Nonetheless, evidence for widespread transmission within deer farms is equivocal. In a study of four captive cervid facilities in Texas and Alabama, only one WTD herd displayed a seroprevalence > 90% for SARS-CoV-2 [24,25]. There was no evidence of infection among the herds at the other three facilities. With the uncertainty of SARS-CoV-2 transmission dynamics in farmed deer facilities, it is imperative to understand how the human–deer interface may influence SARS-CoV-2 transmission.

In this study, we compared the prevalence of SARS-CoV-2 in farmed and free-ranging WTD in Florida to determine if varying levels of human engagement impacted infection or exposure rates. If direct human exposure was a significant route for SARS-CoV-2 transmission in WTD, we would anticipate the following: (1) a higher prevalence of infection among farmed deer that regularly interacted with humans compared with their free-ranging counterparts; (2) farmed WTD to exhibit evidence of exposure to SARS-CoV-2, as indicated by the presence of virus-specific antibodies, and for the prevalence of seropositive animals to be higher in a farmed setting compared with free-ranging deer; and (3) the timing of infection and viral variants present in WTD to align with those observed in humans. Here, we addressed these hypotheses by conducting RT-qPCR and serological assays to assess the infection and exposure status of Florida’s farmed and free-ranging WTD and compared our findings with concurrent trends of SARS-CoV-2 in the human population in Florida.

## 2. Materials and Methods

### 2.1. WTD Specimen Collection

#### 2.1.1. Farmed WTD

Between January 2019 and December 2022, biological specimens were collected from 381 individual farmed white-tailed deer on 56 farms across 25 counties in Florida (Figure 1). All specimens were obtained either through field necropsies or live deer sampling events conducted by farm owners or managers and technicians from the University of Florida Cervidae Health Research Initiative (CHeRI) following approved procedures (IACUC protocols 201508838 and 201609412). Most necropsied samples were collected within 3 to 24 h after death, and none were collected later than 48 h. All biosamples were kept on ice until they were returned to the laboratory, where they were stored at −80 °C. Specimens from white-tailed deer in 2019 were included in our surveillance efforts as an additional way to validate the specificity of the assays. Nasal swabs (N = 2) and lung tissue (N = 15) collected before January 2020 were considered “pre-pandemic”, while specimens (nasal swabs N = 333, lung tissue N = 245) collected from 2020 to 2022 were considered to have been collected during the pandemic (Appendix A). Among the farmed deer samples were 178 paired lung tissue and nasal swabs (Appendix A).

Whole blood specimens were collected during the SARS-CoV-2 pandemic from two deer farms consisting of 352 serial samples from 161 live white-tailed deer in the spring and fall seasons of 2020–2022 (Appendix A). Two hundred pre-pandemic serial samples were collected from 168 individuals on four farms and used for validation in our subsequent surveillance methods. Blood was collected via venipuncture into 8.5 mL Tiger Top serum separator tubes. After 30 min of incubation at room temperature, whole blood was separated into serum by centrifuging at 1500 G for 15 min. For 132 animals, paired nasal swabs were collected at the time of blood sampling. We conducted seven distinct sampling events on the first farm and serum was collected from a total of 119 individual animals, 66 of which had longitudinal sampling (Appendix A). Of these, 27 deer were sampled twice, 8 were sampled three times, 13 were sampled four times, 9 were sampled five times, 7 were sampled six times, and 2 were sampled at all seven events. For the second farm, serum samples were obtained during three sampling events. Among the 44 individual deer sampled from the second farmed deer herd, 4 were sampled on two separate occasions, while 10 were sampled three times. The status of SARS-CoV-2 infection in farmers during sample collection was unknown; however, we describe local county-wide human infection rates during the time of sampling as a proxy for the probability of human infection.

#### 2.1.2. Free-Ranging WTD

Specimens were also obtained from free-ranging white-tailed deer in 26 northern Florida counties from January 2019 to January 2023 (Figure 1). Prior to the pandemic in 2019, 175 retropharyngeal lymph node tissues were collected. During the pandemic, specimens included lymph node tissues (N = 491), nasal swabs (N = 33), and serum samples (N = 29) (Appendix A). Lymph node tissue and individual nasal swabs (N = 4) were acquired through either hunter- or agency-harvest and were collected for unrelated research objectives. Serum specimens and their paired nasal swabs (N = 29) were initially collected from live deer while under sedation for agency-related research purposes. These specimens were obtained from deer in Putnam County (N = 20) and Baker County (N = 9). Sampling was opportunistic and limited to animals from northern Florida, where the majority of farmed WTD specimens were collected. This sampling scheme facilitated a regional comparison of SARS-CoV-2 prevalence between northern Florida farmed and free-ranging WTD.

### 2.2. RNA Extraction and Reverse Transcription Quantitative PCR (RT-qPCR)

RNA was extracted from material extruded from nasal swabs, lung tissue, and retropharyngeal lymph nodes using the Qiagen RNeasy Mini Kit (Qiagen, Germantown, MD, USA) following the manufacturer’s protocol. Detection of SARS-CoV-2 RNA was performed based on the Real-Time RT-PCR protocol established by the Centers for Disease Control and Prevention (CDC) with minor modifications [26]. Each 10 μL reaction included 2.5 μL of the extracted sample RNA and 7.5 μL of prepared AgPath-ID™ One-Step RT-PCR master mix (Applied Biosystems, Foster City, CA, USA) containing SARS-CoV-2 nucleocapsid (N1)-specific primer and probes (Integrated DNA Technologies, Coralville, IA, USA). The master mix also included a separate exogenous internal positive control (Applied Biosystems) to differentiate true negatives from potential PCR inhibition. A 5-point reference standard curve of a quantified positive control (Integrated DNA Technologies), ranging from 1–1000 genome copies/μL, was run on each 96-well plate. Each sample was run in duplicate, and a positive C_q_ cutoff value was set at 38 cycles. If a sample tested positive in at least one of the two wells using the N1 primer, it was retested using SARS-CoV-2 N2-specific primer and probes to confirm the positive result. To assess potential tissue tropism, we tested additional tissues, including lung, heart, spleen, kidney, and liver tissues, using the same RT-qPCR protocol for each SARS-CoV-2 positive farmed deer.

### 2.3. Genomic Sequencing and Lineage Classification

Because the RT-qPCR reactions indicated that the amount of virus RNA recovered from the specimens of the SARS-CoV-2-positive WTD was low (Cq 26) and thus unsuitable for common next-generation sequencing approaches, for which a Cq ≤ 20 is optimal, Sanger sequencing based on a gene-walking approach with nonoverlapping primers was used to obtain the virus genomic sequences [27]. cDNA was produced using AccuScript high-fidelity reverse transcriptase (Agilent Technologies, Santa Clara, CA, USA) using primers based on SARS-CoV-2 genome sequences that had been posted in GISAID (https://www.gisaid.org/, accessed on 21 April 2024) early during the outbreak. The resulting cDNAs were PCR-amplified with Q5 polymerase (New England BioLabs, Ipswich, MA, USA) and specific primers, and the 5′ and 3′ ends of the SARS-CoV-2 genome were determined using a Rapid Amplification of cDNA Ends (RACE) kit (Life Technologies, Inc., Carlsbad, CA, USA). Sequencing was attempted on specimens that tested positive for SARS-CoV-2 from both farmed (N = 3) and free-ranging (N = 4) WTD. However, viral genomic sequences were only obtained from the RNA recovered from the farmed WTD.

### 2.4. SARS-CoV-2 Antibody Detection

Serum specimens collected from farmed (N = 352) and free-ranging (N = 29) WTD between 2020 and 2022 were tested for the presence of SARS-CoV-2-neutralizing antibodies. WTD are susceptible to infection by other coronaviruses [28], and antibody testing for SARS-CoV-2 can be confounded by cross-reactivity with non-target coronaviruses. To address this issue, a positive interpretation was made only if a sample tested positive on three serological assays, which included an in-house indirect ELISA, a commercial SARS-CoV-2 surrogate Virus Neutralizing Test (sVNT) (Genscript, Piscataway, NJ, USA), and a serum neutralization assay.

#### 2.4.1. SARS-CoV-2 Antibody Detection by Indirect ELISA

An in-house indirect ELISA was used to assess antibodies specific to the receptor-binding domain (RBD) and nucleocapsid (N) viral proteins for both the Alpha and Delta (B.1.617.2) variants following a previously established protocol [7,29]. Wells were coated with 100 ng of Alpha RBD (Genscript), Alpha N (Genscript), Delta RBD (Genscript), or Delta N (Genscript) protein in 100 μL of carbonate–bicarbonate coating buffer (Sigma Aldrich, Burlington, MA, USA) and incubated at 4 °C overnight. Following incubation, the plates were washed three times with Millipore phosphate-buffered saline–TWEEN 20 (PBS-T) (Sigma Aldrich). The plates were then blocked with 200 μL per well of casein-blocking buffer (Sigma Aldrich), incubated at room temperature for one hour, and washed three times with PBS-T. Serum samples were diluted 1:400 using the casein blocking buffer, and 100 μL of each diluted sample was added to each well of the ELISA plate. Plates were then incubated again for one hour at room temperature and washed three times with PBS-T. A 100 μL solution of horse-radish peroxidase (HRP)-labeled Rabbit Anti-Deer IgG (H + L) secondary antibody (VWR, Radnor, PA, USA) (diluted 1:1000 or 100 ng/mL) in casein blocking buffer was then added to each well and incubated for one hour at room temperature. After incubating again, the plates were washed five times with PBS-T, and then 100 μL of TMB ELISA substrate solution (Abcam, Waltham, MA, USA) was added to all wells. The plates were then covered and incubated at room temperature for 5 min before stopping the reaction with 100 μL of stop solution (Abcam). The optical density (OD) of each ELISA plate was read at 450 nm. On each plate, serum collected from a deer experimentally infected with SARS-CoV-2 during an animal trial study conducted at Kansas State University [7] served as a positive control. Additional serum collected from farmed deer from 2016 to 2019 (N = 200) was also subjected to indirect ELISA and used to establish a positive cutoff value. All serum samples collected before January 2020 were considered pre-COVID/SARS-CoV-2 seronegative and used to determine the cutoff value as the average OD of the negative serum + 4X the standard deviation. Samples with ODs above the cutoff were considered positive.

#### 2.4.2. SARS-CoV-2 Antibody Detection by sVNT

All serum collected during the time of the SARS-CoV-2 pandemic (2020–2022) from both farmed (N = 352) and free-ranging deer (N = 29), along with a subset of randomly chosen pre-covid/SARS-CoV-2 seronegative samples (2016–2019) (N = 125), was tested using a commercially available SARS-CoV-2 surrogate virus neutralizing test (sVNT) following the manufacturer’s protocol (Genscript). Each plate included the positive control obtained from Kansas State University, as well as the kit-provided positive and negative controls. Samples were run in duplicate and were considered positive if their percent inhibition exceeded 30%. While this assay has not been validated specifically for detecting SARS-CoV-2-specific antibodies in deer, positive results have been interpreted as detecting SARS-CoV-2 antibodies in other serological studies involving WTD [1,12].

#### 2.4.3. Virus Neutralizing Antibodies

We validated all sVNT and ELISA-positive samples (N = 10) and three seronegative controls using a virus-neutralizing antibody assay. All VNT work was performed in a BSL3 laboratory. SARS-CoV-2 neutralizing antibodies in sera were quantified using a microneutralization assay as previously described [7,29]. Heat-inactivated (56 °C/30 min) serum samples were subjected to 2-fold serial dilutions starting at 1:8 and tested in duplicate. SARS-CoV-2 virus stocks were diluted to 100 TCID_50_ of SARS-CoV-2 virus in 100 μL DMEM culture media and added 1:1 to 100 μL of the sera dilutions and incubated for 1 h at 37 °C. The mixture was subsequently cultured on Vero-E6/TMPRSS2 cells in 96-well plates. The neutralizing antibody titer was recorded as the highest serum dilution at which at least one of the wells showed complete virus neutralization based on the absence of CPE observed under a light microscope at 96 h post-infection. Separate assays were performed to determine virus neutralization against the Wuhan-like (USA/WA1/2020) and Delta variant (hCoV-19/USA/NYMSHPSP-PV29995/2021; lineage B.1.617.2) strains of SARS-CoV-2. Although our sample collection occurred through the peak of Omicron, we chose to conduct neutralization assays against earlier variants. Previous studies have suggested that antibodies induced by Omicron exhibit low cross-neutralizing responses against older variants in humans [30]; however, there is no evidence indicating that this also occurs within WTD. Serological studies of WTD samples collected during Omicron showed that these samples could neutralize earlier variants, including the Wuhan-like and Delta strains [5,18]. To address the uncertainties in sensitivity, we used three different assays to increase the likelihood of detecting any SARS-CoV-2 variant.

### 2.5. Data Analyses

#### 2.5.1. Statistical Analysis

Agresti–Coull confidence intervals (CIs) were calculated to estimate the prevalence of SARS-CoV-2 infection and evidence of exposure in Florida’s farmed and free-ranging WTD. Prevalence of SARS-CoV-2 infection and exposure were considered significantly different between the sampled farmed and free-ranging populations when confidence intervals did not overlap. Overlapping intervals indicated that the prevalence estimates were not significantly different.

#### 2.5.2. Human Data Source

To understand how the patterns of SARS-CoV-2 in Florida’s deer compared with those in the human population, we collected human disease data from multiple sources. We obtained the total count of human COVID-19 cases in counties where SARS-CoV-2-positive WTD were found from the Florida Department of Health [31] and incorporated the information into a timeline demonstrating the total number of COVID-19 cases in humans from those counties. SARS-CoV-2 Pango lineage classifications were also utilized for all available Florida human SARS-CoV-2 isolates from 2020 to 2022 sourced from the National Center for Biotechnology Information (NCBI) Genbank (N = 188,264) to visualize the proportion of viral lineages in Florida’s human population. This allowed us to compare the timing of infection and viral lineages in WTD with those present in the human population.

#### 2.5.3. Phylogenetic Analysis

Forty-five whole genome sequences of SARS-CoV-2 isolated from humans in Florida between 2020 and 2022, along with three complete sequences from farmed Florida WTD, were used to construct a maximum likelihood (ML) phylogenetic tree. SARS-CoV-2 sequences from humans, used within our phylogenetic tree, were sourced from either the NCBI GenBank or the Global Initiative on Sharing All Influenza Data (GISAID) databases. Each sequence underwent alignment by amino acid translation using Geneious Prime (Version 2023.2.1) with the MAFFT algorithm [32]. Subsequently, the maximum likelihood phylogenetic tree was created using IQ-TREE model GTR+F+R2 [33].

## 3. Results

### 3.1. Prevalence of SARS-CoV-2 Infection in Farmed and Free-Ranging WTD

Among the 191 pre-pandemic specimens (collected prior to December 2019), we did not find any SARS-CoV-2 RNA-positive cases in either farmed or free-ranging WTD. All downstream analyses of RNA prevalence include only data from individuals sampled during the pandemic (January 2020–December 2022). Among the farmed deer sampled during the pandemic, SARS-CoV-2 RNA was found in 3 of 381 individuals (0.78%, 95% CI: 0.15–2.40%). All three of these individuals were located on the same farm; therefore, of the 56 farms sampled, only one farm had SARS-CoV-2-positive deer at the time of sampling.

Only the nasal swabs from each positive farmed deer contained SARS-CoV-2 RNA; other tested tissues from these animals, including lung, heart, spleen, kidney, and liver tissues, did not indicate the presence of the virus. The positive deer (CHeRI identification: OV1519, OV1520, and OV1562) were all located on the same farm in a northern Florida county (Figure 1). From July to December 2021, amid an apparent outbreak of epizootic hemorrhagic disease (EHDV), a total of 20 deceased WTD were necropsied on the farm, with 70% testing positive for EHDV viral RNA and showing clinical signs of hemorrhagic disease. Within this cohort, three animals tested positive for SARS-CoV-2 infection, resulting in a prevalence of 15.0% (95% CI: 4.39–36.88%). The positive deer were male (OV1562) and two female fawns (OV1519 and OV1520), all under three months old. OV1519 and OV1520 were found to be coinfected with epizootic hemorrhagic disease virus (EHDV) and exhibited clinical signs consistent with EHDV infection, such as congested and hemorrhagic lungs and kidneys, along with potential pneumonia, with organ failure likely as the cause of death. These clinical signs were typical of animals infected with EHDV. OV1562 was euthanized by the farm owner; however, the animal tested negative for EHDV and exhibited only slight congestion in the kidneys and an insignificant amount of mucus in the internal bronchi. All major organs appeared normal, and the cause of clinical illness was deemed unknown.

The prevalence of SARS-CoV-2 infection in free-ranging deer collected during the pandemic was 0.76% (4 out of 524; 95% CI: 0.22–2.02%). SARS-CoV-2 RNA was found in the lymph nodes of four free-ranging deer located in different counties at various times between 2021–2022 (Figure 1 and Figure 2). The earliest detection of a SARS-CoV-2-positive deer was a female yearling in Clay County, Florida, harvested by a hunter in March 2021. Subsequently, three additional positive cases were identified in adult males from Putnam, Wakulla, and Suwannee Counties in June, July, and October of 2022.

### 3.2. Little Evidence of SARS-CoV-2 Seropositivity in Florida WTD

The results of three serological tests conducted to detect SARS-CoV-2-specific neutralizing antibodies showed limited evidence of SARS-CoV-2 seroprevalence in both farmed and free-ranging WTD in Florida. Out of 381 serum specimens from 190 individual farmed and free-ranging WTD collected during the pandemic, five individuals tested positive for at least one assay (1.31%, 95 CI: 0.47–3.12%). Serum from six individual deer tested positive for at least one of the RBD proteins (Alpha or Delta) used in the in-house ELISA assay, including a pre-covid sample collected in 2018 (Table 1). None of the serum tested positive for antibodies against either Alpha or Delta N proteins. Only one sample tested positive by the commercial sVNT, which was serum collected from a free-ranging WTD in 2022. The serum collected from this free-ranging deer demonstrated 89.6% inhibition of neutralizing antibodies against the binding of the RBD protein.

Among the subset of samples subjected to microneutralization testing, virus-neutralizing antibodies against both the Wuhan-like (USA/WA1/2020) and Delta variant (hCoV-19/USA/NYMSHPSP-PV29995/2021; lineage B.1.617.2) strains of SARS-CoV-2 were detected in one free-ranging WTD sampled in 2022, which previously tested positive using both the commercial sVNT and the Alpha and Delta RBD proteins within the ELISA assay. The neutralizing titers ranged from 1:32 when tested against WA1 to 1:64 when tested against Delta. All other samples showed titers < 1:8 and were considered negative for SARS-CoV-2 virus-neutralizing antibodies. The serum specimens collected from a free-ranging WTD (WOV43) in November 2022 were the only specimens considered seropositive for SARS-CoV-2-specific and virus-neutralizing antibodies by all three serological assays. This deer was an adult male located in Putnam County (Figure 1).

### 3.3. Comparison of SARS-CoV-2 Prevalence in Farmed and Free-Ranging WTD in Northern Florida

We found no significant difference in the prevalence of SARS-CoV-2 between farmed and free-ranging WTD in northern Florida. In these counties, the prevalence of SARS-CoV-2 among farmed WTD was 0.85% (3 out of 351 deer; 95% CI: 0.17–2.60), while free-ranging WTD had a prevalence of 0.76% (4 out of 524 deer; 95% CI: 0.22–2.02%). The overlapping confidence intervals for infection prevalence in both populations suggest that the prevalence of SARS-CoV-2 infection in farmed WTD was not significantly different from that in free-ranging WTD.

We found no statistical difference in SARS-CoV-2 virus-neutralizing antibodies among farmed and free-ranging WTD; 3.44% for free-ranging deer (1 out of 29 deer; 95% CI: −0.84–18.63%) vs. 0% (0 out of 352, 95% CI: −0.22–1.30%) for farmed deer. The overlapping confidence intervals between farmed and free-ranging WTD suggest no significant difference in seroprevalence between the two populations.

### 3.4. Timing of Infection and SARS-CoV-2 Variants in Florida’s Humans and WTD

The timing of infection and variant similarity observed in the SARS-CoV-2-positive WTD closely mirrored the trends seen in Florida’s human population. Both farmed and free-ranging deer were infected with SARS-CoV-2 concurrently with the human population (Figure 3). Instances of positive deer cases occurred in the same counties and when the total number of human COVID-19 cases peaked. Additionally, during the period when the Delta variant was prevalent among humans, it was also the variant detected in farmed WTD (Figure 3).

Efforts were made to sequence the entire genomes of the SARS-CoV-2 isolates from both the positive nasal swabs of farmed WTD (N = 3) and the positive retropharyngeal lymph node tissues of free-ranging deer (N = 4). We were unable to obtain whole genome sequences for the viruses isolated from any free-ranging WTD. This challenge likely stemmed from each sample having a high cycle threshold (C_q_) value (>32) from RT-qPCR, indicating insufficient genetic material required for whole genome sequencing. However, we were able to successfully obtain genomic sequences for the viruses isolated from farmed WTD OV1519, OV1520, and OV1562.

Compared with the reference strain (GenBank no. NC_045512.2), the complete SARS-CoV-2 genome in OV1520 (GenBank accession # PP725203.1) had amino acid substitutions: Spike D614G, Spike D950N, Spike E156G, Spike F157del, Spike G142D, Spike L452R, Spike P681R, Spike R158del, Spike T19R, Spike T478K, M I82T, N D63G, N D377Y, N G215C, N R203M, NS3 G76S, NS3 S26L, NS7a T120I, NS7a V82A, NS7b T40I, NSP3 A488S, NSP3 P1228L, NSP3 P1469S, NSP4 T492I, NSP4 V167L, NSP6 A2V, NSP6 T77A, NSP12 G671S, NSP12 P323L, NSP12 T225I, NSP13 P77L, NSP13 S350L, NSP14 A394V. These mutations indicate that the virus was a member of GISAID Clade GK, Pango Lineage AY.46.6 (Pango v.4.1.1 PLEARN-v1.12), Delta (B.1.617.2-like) (Figure 3) [34]. Unlike the complete genome sequence of the virus from OV1520, we did not obtain the complete genome sequence of the virus in OV1519 (GenBank accession # PP724850.1). Instead, seven sections with a total of 16,450 nucleotides were determined (data provided upon request). These sequences had 100% identities with the corresponding sequences in the SARS-CoV-2 genome isolated from OV1520. Additionally, a complete SARS-CoV-2 genome was obtained for OV1562 (GenBank accession # PP725253.1), and it had the following amino acid substitutions: Spike D614G, Spike E156G, Spike F157del, Spike G142D, Spike L452R, Spike P681R, Spike R158del, Spike T19R, Spike T478K, M I82T, N D63G, N D377Y, N G215C, N R203M, NS3 S26L, NS7a T120I, NS7a V82A, NS7b T40I, NSP3 A488S, NSP3 K1302T, NSP3 P1044S, NSP3 P1228L, NSP3 P1469S, NSP4 T492I, NSP4 V167L, NSP6 A2V, NSP6 I162V, NSP6 T77A, NSP12 G671S, NSP12 P323L, NSP13 P77L, NSP14 A394V. The mutations indicated that the virus was a member of GISAID Clade GK, Pango Lineage AY.46.6 (Pango v.4.1.1 PLEARN-v1.12), Delta (B.1.617.2-like), differing from the genome of OV1520 by four nucleotides.

When compared with human isolates representing each of the major SARS-CoV-2 variants present within the state of Florida, all sequenced farmed WTD samples clustered together within the Delta clade (Figure 4). Each of the sequenced isolates from the farmed WTD shared > 90% similarity with SARS-CoV-2 sequences obtained from humans in Florida, Mississippi, and North Carolina, dating between June and September 2021.

## 4. Discussion

Despite concerns regarding the potential for increased prevalence of SARS-CoV-2 in farmed WTD compared with free-ranging deer, our study revealed no significant difference in the prevalence of infection or exposure between farmed and free-ranging populations in northern Florida. Nonetheless, among the small subset of WTD that did test positive for SARS-CoV-2, the timing of their infections and the similarity of their viral sequences to those found in humans provides evidence for the occurrence of spillover from humans to deer, albeit less frequently than anticipated based on studies from other regions of North America.

At the time of our study, a large portion of the human population within these counties was susceptible to SARS-CoV-2 infection. In the sampled northern Florida counties, the average rate of human vaccination against SARS-CoV-2 by the end of 2022 was 54.64% [35]. Considering the elevated risk of infection in humans and the potential for spillover, we expected to detect a non-zero prevalence of SARS-CoV-2 in Florida’s WTD, specifically within farmed animals that were likely exposed to the virus via owners and their families or farm managers and workers. However, in both farmed and free-ranging populations, SARS-CoV-2 infection prevalence was less than 1%.

The level of exposure to SARS-CoV-2 in farmed and free-ranging WTD did not align with our expectations. Our serological sampling approach for farmed WTD was unique in that we collected repeated samples from herds at two farming facilities. Since SARS-CoV-2-specific antibodies persist for at least 13 months in naturally infected WTD [36], our serological approach would have enabled us to monitor potential virus exposure throughout the three-year pandemic sampling period. However, our findings showed limited evidence of SARS-CoV-2 exposure among any of Florida’s WTD, with seroprevalence rates of 0% and 3.44% of the farmed and free-ranging deer sampled, respectively. Even with the perceived elevated risk of exposure for farmed WTD [37], no seropositive animals were identified on either of the two farms that were repeatedly sampled. The only individual that tested seropositive for SARS-CoV-2 neutralizing antibodies was a single free-ranging deer whose location at the time of sampling was in a peri-urban area of north central Florida.

Our seroprevalence findings in Florida’s WTD contradict our expectation that farmed WTD, if human–deer transmission played a significant role in spillover, would exhibit higher SARS-CoV-2 prevalence compared with free-ranging deer.

Even though seroprevalence in WTD was lower than we expected, the timing of infection and the viral sequences isolated from farmed WTD provided strong evidence for human-to-deer spillover. Positive SARS-CoV-2 cases were observed in both farmed and free-ranging WTD during the peaks in Florida’s human cases, implying a similarity between infections in deer and humans. The viral sequences isolated from infected farmed WTD in Florida shared > 90% genetic similarity with those found in humans. The similarity among viral strains collected from humans and deer parallels findings from other regions within the United States where viruses detected in humans were also identified in WTD, suggesting spillover in some capacity [14]. Together, these findings suggest that spillover from humans to WTD likely occurs; however, they do not indicate a specific mode of transmission. Based on our results, direct contact from humans to deer was not a common route of transmission among Florida-farmed deer.

To date, this study provides the first estimates of SARS-CoV-2 prevalence in both farmed and free-ranging WTD in the southeastern United States. In our study, the low prevalence of infections in Florida’s free-ranging WTD (<1%), along with low seroprevalence rates, sharply contrasts with estimates of SARS-CoV-2 prevalence in free-ranging deer in other regions of North America [1,2,6,9,10,11,12,13,14,15,16,17,18,19]. SARS-CoV-2 viral RNA has been reported in free-ranging WTD in 29 states across the United States, with prevalences ranging from 0.6% in one New York deer population (September–December 2020) [17] to 36% in one Ohio deer population (January–March 2021) [9]. The virus lineages reported in WTD thus far are diverse and include variants that were concurrently circulating within the local human populations [14,16,17], a similar trend seen with Florida’s positive-farmed deer. Similarly, seroprevalence in other North American free-ranging WTD has been as high as 67% in Mississippi from 2021–2022 [18]. Differences in the prevalence of SARS-CoV-2 in free-ranging WTD may be attributed to different environmental factors, such as the potential that an intermediate host may be facilitating spillover, or it could be attributed to cross-reactivity of serological tests that inflate serological prevalence. Further surveillance of SARS-CoV-2 in wildlife populations using stringent methodologies is necessary to fully understand the transmission dynamics of the virus.

The low prevalence of SARS-CoV-2 in Florida’s farmed WTD is not unprecedented. There have been two additional studies performed on farmed cervid herds in the United States [24,25], both of which demonstrated low occurrence of SARS-CoV-2 seroprevalence or infection in four out of the five farming facilities sampled during the pandemic. In Alabama, a university-owned herd suspected of SARS-CoV-2 exposure through infected faculty and student handlers showed no evidence of infection or seroprevalence within their WTD between October 2019 and January 2022 [24]. Similarly, within three captive cervid facilities in Texas, only one WTD herd exhibited high seroprevalence (94.4%), indicating exposure and dissemination of SARS-CoV-2 in only one of the tested herds [25]. The findings from captive facilities, including deer farms in Florida, support the hypothesis that direct transmission between humans and deer may occur less frequently than expected, given the close contact between humans and deer at these facilities, and that other routes of transmission may be at play.

While our study provides valuable information on the prevalence of SARS-CoV-2 in Florida WTD, it is important to consider the limitations of our study design. Our sampling method relied on convenience sampling, resulting in most of our specimens originating from northern Florida. Consequently, our results may not fully represent the entirety of Florida’s farmed and free-ranging WTD populations. It is possible that SARS-CoV-2 prevalence in WTD in central and southern Florida could differ from our current findings. Additionally, the timing of our specimen collection for necropsied and harvested animals might have led to missed infections. Experimental infection studies have shown that WTD typically shed SARS-CoV-2 for approximately 5–10 days post-inoculation [7,8]. Therefore, our sampling timing may not have captured all infected individuals. Our serology sampling locations for both farmed and free-ranging WTD were limited to only two farms and two counties. Despite this, given the number of serum specimens collected and the unique repeated sampling throughout the pandemic, we would have still expected to detect some seropositive animals to demonstrate exposure, particularly among farmed deer. While some limitations are evident within our study design, our overall conclusions are well-supported by our results. We found that SARS-CoV-2 was not prevalent among farmed and free-ranging WTD in northern Florida. Future studies that employ more comprehensive sampling methods across different regions of Florida could provide a better understanding of SARS-CoV-2 prevalence in WTD populations statewide.

The ongoing evidence of SARS-CoV-2 spillover from humans to other mammalian species raises significant concerns regarding the potential for anthropozoonosis. WTD, especially those in farmed facilities with frequent human interaction, are particularly worrisome due to their demonstrated susceptibility to infection. Our comparison of SARS-CoV-2 prevalence in Florida’s farmed and free-ranging WTD revealed minimal indication of exposure in these populations. These findings highlight the importance of further research to understand the ecology of SARS-CoV-2 in wildlife species and to identify key transmission routes for spillover events.

## Figures and Tables

**Figure 1 viruses-16-01886-f001:**
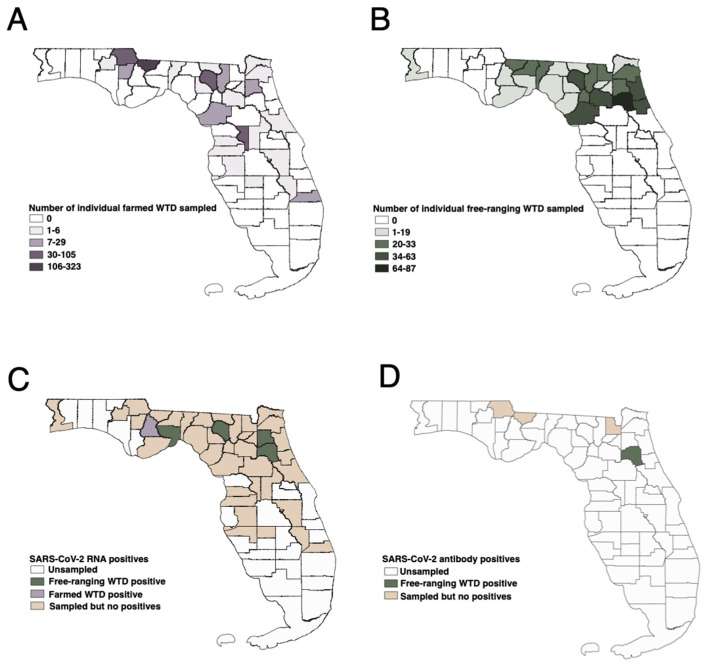
Geographic distribution of WTD sampling efforts from 2020 to 2022 and SARS-CoV-2-positive cases. (**A**) Total number of farmed WTD specimens by county. (**B**) Total number of free-ranging WTD specimens by county. (**C**) Counties with SARS-CoV-2 RNA positive cases as determined by RT-PCR. (**D**) Counties with SARS-CoV-2 neutralizing antibodies positive WTD.

**Figure 2 viruses-16-01886-f002:**
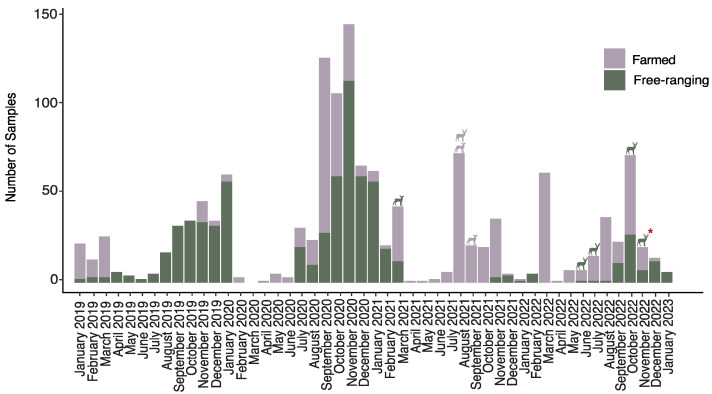
Temporal distribution of all specimens from farmed and free-ranging WTD and the timing of SARS-CoV-2 positives for infection or virus-neutralizing antibodies. WTD status is color-coded, with sex distinctions. An asterisk denotes the virus-neutralizing antibody-positive deer, while all others were positive for viral infection by RT-qPCR.

**Figure 3 viruses-16-01886-f003:**
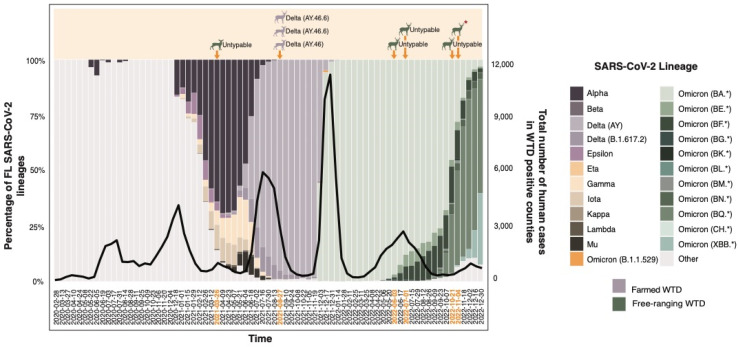
The temporal distribution of SARS-CoV-2 lineages in Florida humans and WTD from 2020 to 2022. The black line depicts the cumulative human cases in counties with confirmed positive WTD. Positioned above the graph, designated time points mark instances of positive WTD alongside the discerned SARS-CoV-2 lineages determined through whole genome sequencing. An asterisk denotes the virus-neutralizing antibody-positive WTD, while all others were positive for SARS-CoV-2 infection by RT-qPCR.

**Figure 4 viruses-16-01886-f004:**
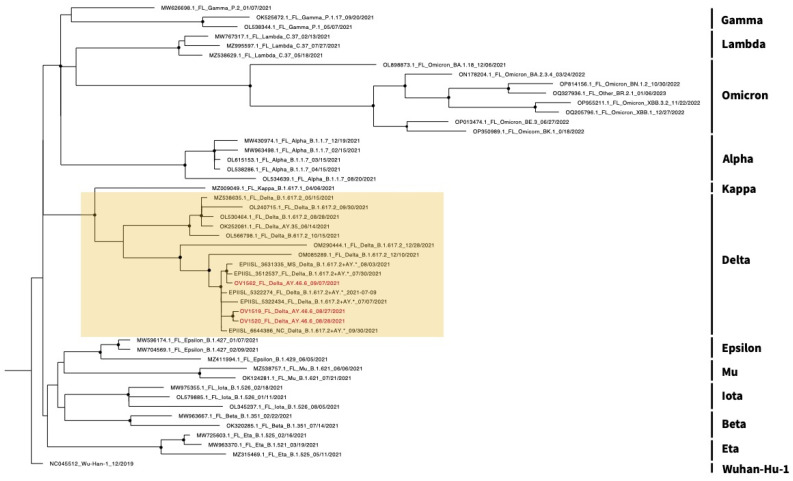
Maximum likelihood cladogram of the relationships of the whole genome SARS-CoV-2 nucleotide sequences isolated from Florida humans and farmed WTD. The SARS-CoV-2 isolates include their accession references from either NCBI or GISAID, location, WHO variant name, Pangolin lineage, and collection date. Nodes with black circles are supported by bootstrap values of >90%. The tree was rooted with the Wuhan-Hu-1 reference strain. The farmed WTD isolates are in red, and the Delta clad is highlighted in yellow.

**Table 1 viruses-16-01886-t001:** Serological evidence of SARS-CoV-2 exposure in Florida WTD. Results of individual WTD that tested positive (+) for at least one serological assay, including the in-house ELISA using RBD and N proteins from both the Wuhan-like and Delta variants of concern. Negative (−) results for these individuals were also included.

ID	County	Range Status	Sampling Time		In-House ELISA			Genscript sVNT	VNT
				⍺ RBD	Δ RBD	⍺ N	Δ N	% Inhibition	Titer Ratio
OV_71	Gadsden, FL	Farmed	24 April 2018	+	+	−	−	0	<1:8
OV_1551	Gadsden, FL	Farmed	30 August 2021	−	+	−	−	0	<1:8
OV_709	Gadsden, FL	Farmed	21 March 2022	+	+	−	−	0	<1:8
OV_1746	Jackson, FL	Farmed	22 October 2022	−	+	−	−	0	<1:8
WOV_43	Putnam, FL	Free-ranging	14 November 2022	+	+	−	−	89.60	1:32 WA1, 1:64 Delta
WOV_56	Putnam, FL	Free-ranging	4 January 2023	+	+	−	−	0	<1:8

## Data Availability

All data are available within the main text or in the Appendix A.

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
