# Peer review of "Low Prevalence of SARS-CoV-2 in Farmed and Free-Ranging White-Tailed Deer in Florida"

_viruses, 2024, doi:10.3390/v16121886_

Round 1
Reviewer 1 Report
Comments and Suggestions for Authors
Savannah G. Grace et al.
Low prevalence of SARS-CoV-2 in farmed and free-ranging white-tailed deer in Florida
Comments:
The authors surveyed SARs-COV-2 in WTD of Florida. They did an excellent job approaching this research topic and presented their findings in a manner worthy of publication. The paper is well written and referenced appropriately.
I have a few recommendations:
(1) Figure 3 should be bigger. It is hard to read as this size. This may be an issue for the editor to fix. Landscape version may help. Also, the Y-axis title is crowded in the font chosen.
(2) Figure 4 is also small and should be enlarged. Maybe image 3 and 4 should be flipped in the landscape position.
(3) Throughout the method section the Authors used the catalog # for the reagents used. I recommend changing that to reflect only the company and their location. Catalog #’s may change over time.
Example: Line 147 “RNA was extracted from material extruded from nasal swabs, lung tissue, and retropharyngeal lymph nodes using the Qiagen RNeasy Mini Kit (Qiagen, catalog# 74106)”
Should be (Qiagen, Hilden, Germany).
Author Response
Comments:
The authors surveyed SARs-COV-2 in WTD of Florida. They did an excellent job approaching this research topic and presented their findings in a manner worthy of publication. The paper is well written and referenced appropriately.
I have a few recommendations:
(1) Figure 3 should be bigger. It is hard to read as this size. This may be an issue for the editor to fix. Landscape version may help. Also, the Y-axis title is crowded in the font chosen.
Response: Iagree. I will work with the managing editor to see if we can get the typeset pdf copy to set Figures 3 and 4 in landscape orientation.
(2) Figure 4 is also small and should be enlarged. Maybe image 3 and 4 should be flipped in the landscape position.
Response: I agree. I will work with the managing editor to see if we can get the typeset pdf copy to set Figures 3 and 4 in landscape orientation.
(3) Throughout the method section the Authors used the catalog # for the reagents used. I recommend changing that to reflect only the company and their location. Catalog #’s may change over time.
Example: Line 147 “RNA was extracted from material extruded from nasal swabs, lung tissue, and retropharyngeal lymph nodes using the Qiagen RNeasy Mini Kit (Qiagen, catalog# 74106)”
Should be (Qiagen, Hilden, Germany).
Response: We have edited all manufacturer details. We have omitted the catalog number and included the city, state and country.
Reviewer 2 Report
Comments and Suggestions for Authors
Pathogenic coronaviruses that have emerged over the past two decades have demonstrated a clear capacity for zoonotic transmission. Indeed, SARS-CoV-1, which emerged early in this century, is commonly accepted to have spread from bats to palm civets to humans. MERS-CoV, which emerged in 2012, is thought to have spread from dromedary camels to humans. Most impactful, of course, is SAR-CoV-2 (COVID-19), which emerged in China in late 2019 and spread worldwide, causing a worldwide pandemic that claimed millions of lives and caused ongoing pain and suffering. The SARS-CoV-2 virus is known to spread relatively easily between humans and many animal species, both domestic and wild, depending on the conservation of the ACE2 receptor utilized by the virus.
This manuscript describes the findings from a study designed to understand the prevalence of the SARS-CoV-2 virus in farmed and wild white-tailed deer in the state of Florida with the goal of evaluating the potential for zoonotic transmission between the humans and deer, which would be particularly concerning, given the relatively close interaction between the two populations in nature. Samples were taken from sufficiently large groups of both farmed and free-ranging deer over a large area of the state and tested by RT-qPCR for infection and serological assays for exposure to SARS-CoV-2. Fortunately, virus infection was detected in <1% of both deer populations. Similarly, only a single free-ranging deer and no farmed deer at all, tested positive for virus-specific antibodies. Even though the data confirm that the sequences of viruses in the deer and human populations match, this strongly suggests that the risk for zoonotic transmission of SARS-CoV-2 between white-tailed deer and the human population in Florida is extremely low. Although the situation requires constant monitoring, these findings are considered encouraging, especially since the virus has been detected often at much higher prevalence in these deer populations in nearly thirty states.
The study is well-designed and evaluated and the data are appropriately analyzed and interpreted. Limitations of the study are also fully acknowledged, including its primary focus being northern Florida, the timing of sampling and the serological studies being limited to only two farms and counties. There are no experimental or scientific weaknesses or even minor problems in the study. However, although the findings are considered positive with respect to the safety of the human population, the impact of the study is considered quite low.
Author Response
Thank you for your thorough reading of our manuscript. We agree that this manuscript is one in a series of research studies that incrementally adds to our understanding of why SARS CoV2 spillsover to some wildlife populations and not others.